# Emergent Role of IFITM1/3 towards Splicing Factor (SRSF1) and Antigen-Presenting Molecule (HLA-B) in Cervical Cancer

**DOI:** 10.3390/biom12081090

**Published:** 2022-08-08

**Authors:** Maria Gómez-Herranz, Jakub Faktor, Marcos Yébenes Mayordomo, Magdalena Pilch, Marta Nekulova, Lenka Hernychova, Kathryn L. Ball, Borivoj Vojtesek, Ted R. Hupp, Sachin Kote

**Affiliations:** 1Institute of Genetics and Cancer, University of Edinburgh, Edinburgh EH4 2XU, UK; 2International Centre for Cancer Vaccine Science, University of Gdańsk, 80-822 Gdańsk, Poland; 3Masaryk Memorial Cancer Institute, Research Centre for Applied Molecular Oncology, 65653 Brno, Czech Republic

**Keywords:** interferon (IFN), mRNA, ribosome, interferon-induced transmembrane 1 and 3 (IFITM1/3), isoform of serine and arginine-rich splicing factor 1 (SRSF1)

## Abstract

The IFITM restriction factors play a role in cancer cell progression through undefined mechanisms. We investigate new protein–protein interactions for IFITM1/3 in the context of cancer that would shed some light on how IFITM1/3 attenuate the expression of targeted proteins such as HLA-B. SBP-tagged IFITM1 protein was used to identify an association of IFITM1 protein with the SRSF1 splicing factor and transporter of mRNA to the ribosome. Using in situ proximity ligation assays, we confirmed a predominant cytosolic protein–protein association for SRSF1 and IFITM1/3. Accordingly, IFITM1/3 interacted with HLA-B mRNA in response to IFNγ stimulation using RNA–protein proximity ligation assays. In addition, RT-qPCR assays in IFITM1/IFITM3 null cells and wt-SiHa cells indicated that HLA-B gene expression at the mRNA level does not account for lowered HLA-B protein synthesis in response to IFNγ. Complementary, shotgun RNA sequencing did not show major transcript differences between IFITM1/IFITM3 null cells and wt-SiHa cells. Furthermore, ribosome profiling using sucrose gradient sedimentation identified a reduction in 80S ribosomal fraction an IFITM1/IFITM3 null cells compared to wild type. It was partially reverted by IFITM1/3 complementation. Our data link IFITM1/3 proteins to HLA-B mRNA and SRSF1 and, all together, our results begin to elucidate how IFITM1/3 catalyze the synthesis of target proteins. IFITMs are widely studied for their role in inhibiting viruses, and multiple studies have associated IFITMs with cancer progression. Our study has identified new proteins associated with IFITMs which support their role in mediating protein expression; a pivotal function that is highly relevant for viral infection and cancer progression. Our results suggest that IFITM1/3 affect the expression of targeted proteins; among them, we identified HLA-B. Changes in HLA-B expression could impact the presentation and recognition of oncogenic antigens on the cell surface by cytotoxic T cells and, ultimately, limit tumor cell eradication. In addition, the role of IFITMs in mediating protein abundance is relevant, as it has the potential for regulating the expression of viral and oncogenic proteins.

## 1. Introduction

Multiple oncogenic pathways selectively modulate the expression of mRNAs. In addition, the regulation of protein expression is essential for cancer development [1]. Thus, there is a particular scientific interest to unravel which cellular components orchestrate the mRNA transport, as well as the maturation and the translation of specific mRNA products that are associated with tumor progression. Indeed, control of cancer-specific translation is emerging as a new anticancer strategy wherein therapeutic drugs inhibit mRNA translation in a selective manner [2].

Interferons (IFNs) form a family of cytokines that were originally discovered to respond in opposition to the adverse effects of the flu virus; however, subsequent biological roles described for IFNs include activity against cancer development [3]. In this regard, IFNs are used in cancer treatment [4,5,6,7,8,9,10,11] due to their ability to inhibit proliferative pathways, promote cellular apoptosis, and increase the activity of the immune system [12].

Although IFNs show protective effects against tumor progression [13,14,15], a group of IFN-stimulated genes promote tumor development by acquiring resistance to DNA damage, escaping immune surveillance, and increasing metastatic spread [16,17]. These genes are collectively named as IFN-related DNA damage resistance signature (IRDS) and they are expressed upon resistance to radiation and chemotherapy [18,19]. Thus, IFNs can mediate both cancer suppression and growth, depending on the context [20].

There are three protein families described within the mammalian IFN tree: type I (IFNα, IFNβ, IFNe, IFNk, and IFNω), type II (IFNγ), and type III (IFNλ) [21,22]. The core mechanism of action of IFNs involves activating the JAK kinase–STAT signaling pathway that induces the transcription of numerous interferon-stimulated genes (ISGs) [23,24], including IFITM1/2/3.

The immune-related interferon-induced transmembrane (IFITM) protein family is composed of three members: IFITM1, IFITM2, and IFITM3 [25]. IFITM1 is slightly different from IFITM2 and IFITM3, with some studies proposing that IFITM1 is uniquely expressed on the cell surface [26,27]. Only a limited number of interacting partners have been identified for the IFITMs [28,29,30,31]. However, a growing body of research has proven that components of the IFITM family are capable of attenuating the propagation of RNA viruses such as the influenza A virus (IAV), West Nile virus (WNV), dengue virus, severe acute respiratory syndrome (SARS) coronavirus, filoviruses, vesicular stomatitis virus (VSV), and hepatitis C virus (HCV) [32].

In addition to the widely studied antiviral function of the IFITMs, IFITM1 and IFITM3 also function as a pro-oncogenic protein for which the expression has been reported in various cancers such as breast, cervix, colon, leukemia, ovary, brain, and esophagus [18,33,34,35,36,37,38,39,40]. Moreover, IFITM1 is a gene that is expressed upon radiation resistance. Hence, the protective effect described for IFNs against tumors is not well reflected in the role of the IFN-induced IFITM proteins concerning cancer [18,41]. Nonetheless, the molecular mechanism whereby IFITM1 and IFITM3 proteins regulate cancer is not well defined.

Overall, the IFITM functions are associated with blocking viral infection. For instance, IFITMs can reduce viral protein synthesis of human immunodeficiency virus 1 (HIV-1) by preferentially excluding viral mRNA transcripts from translation [42]; thus, IFITMs exhibit an intracellular function linked to suppressed viral propagation. This is consistent with our previous research wherein we found a translational role for the IFITM family in the context of cancer; the deletion of IFITM1/3 genes suppresses the expression of a subset of proteins for which synthesis is mediated by IFNγ. Interestingly, human leukocyte antigen B (HLA-B), which is a component of the IRDS, was significantly identified [43]. This indicates that IFITM1/3 can regulate the synthesis of some antivirals in addition to cancer-associated gene products.

In this report, we further define the molecular implications whereby genetic ablation of IFITM1 and IFITM3 attenuates the synthesis of a subset of IFN-responsive proteins, despite IFITM1/3-knockout cells still retaining the ability to mediate IFN-induced gene expression. Here, we show that IFITM1/3 proteins can associate with the SRSF1 factor, mostly in the cytosol and with HLA-B mRNA. In addition, we identified changes in the 80S ribosomal fraction in the absence of IFITM1/3 expression. Taken together, this all provides additional support for an RNA-binding and/or translational role of the IFITM family of proteins, as reported in response to HIV infection [42].

## 2. Materials and Methods

### 2.1. Cell Culture

The wt-SiHa cells and IFITM1/IFITM3 null cells, which were described previously [43], were grown in RPMI 1640 medium (Invitrogen, Waltham, MA, USA), supplemented with 10% (*v*/*v*) fetal bovine serum (Labtech, East Sussex, UK) and 1% penicillin/streptomycin (Invitrogen, Waltham, MA, USA), and incubated at 37 °C with 5% CO_2_.

### 2.2. Western Blotting

Protein from lysed samples was quantified using the Bradford [44] Protein Assay Dye Reagent (Bio-Rad, Hercules, CA, USA). Proteins were resolved by SDS-PAGE using 15% gels and transferred onto nitrocellulose membranes (Amersham Protran, GE Healthcare, Pittsburgh, PA, USA). Immunoblots were processed by enhanced chemiluminescence (ECL) and quantified as RLU.

### 2.3. Antibodies

Proteins were detected using the following primary antibodies: mouse monoclonal antibodies were generated to a peptide that is identical in IFITM1 and IFITM3, and their characterization was described previously [43]. As this panel of monoclonal antibodies cannot distinguish between the IFITM1 and IFITM3 proteins, the text specifically states that IFITM1 and IFITM3 (shortened to IFITM1/3) proteins were measured when using these antibodies. Other sources of antibodies include mouse monoclonal anti-IFITM2 (Proteintech), rabbit polyclonal anti-SRSF1 (Thermo Fisher Scientific, Waltham, MA, USA), rabbit polyclonal anti-HLA-B (Thermo Fisher Scientific), rabbit polyclonal anti-RPL7a (Cell Signaling Technology, Danvers, MA, USA), mouse monoclonal anti-β-ACTIN (Sigma-Aldrich, St. Louis, MI, USA), and mouse monoclonal anti-GAPDH (Abcam, Cambridge, UK).

### 2.4. Trichloroacetic Acid (TCA) Precipitation

Proteins were precipitated from individual sucrose gradient fractions by TCA precipitation. An amount of 10 μL of 1 μg/μL of bovine serum albumin (BSA) was added to each prechilled eluate, followed by the addition of ice-cold TCA (90 μL), then mixed and incubated on ice for 60 min. The fractions were then centrifuged at 10,000× *g* at 4 °C for 10 min. The pellet was again centrifuged to remove any residual supernatant, washed with 250 μL of ice-cold acetone, and centrifuged at 4 °C for 5 min. Finally, the supernatant was discarded, and a visible small white precipitate was air dried. Protein was then resuspended in 2× sample buffer to proceed with the immunoblot.

### 2.5. Immunofluorescence

The wt-SiHa and IFITM1/IFITM3 null cells were nonstimulated and stimulated with 100 ng/mL IFNγ for 24 h. Cells were fixed with 4% (*v*/*v*) paraformaldehyde in PBS at RT for 15 min, washed with PBS three times, and permeabilized using 0.25% triton X-100 in PBS at RT for 10 min. Then, the cells were again washed with PBS three times and blocked with 3% BSA in PBS for 1 h. The primary antibody was incubated at appropriate dilution (typically 1:1000) overnight at 4 °C. Depending on which host species the primary antibody had been generated, Alexa Fluor 488 goat anti-mouse (Invitrogen), Alexa Fluor 594 donkey anti-rabbit (Invitrogen), or Alexa Fluor 594 goat anti-mouse (Invitrogen) secondary antibody was incubated at RT for 1 h. Coverslips were washed three times with PBS in between each step. Cells were incubated in DAPI (Invitrogen), diluted at 1:10,000 with dH_2_O for 5 min to stain the nucleus. An additional three washes with dH_2_O for 5 min were performed. A single drop of fluorescence mounting medium (S3023, Dako, Glostrup Denmark) was used to mount the cells on the slide. The fluorescent signal was detected using a Zeiss Axioplan 2 microscope (63× or 100× oil immersion objective). Images were acquired using Micro-Manager 1.4 software. Images were processed in ImageJ 2.0 software.

### 2.6. RT-qPCR

The following parameters were considered while designing the primers: Firstly, primers were designed to the codifying region of IFITM1 (F: 5′-ACTGGTATTCGGCTCTGTGAC-3′; R: 5′-GCTGTATCTAGGGGCAGGAC-3′), IFITM3 (F: 5′-CAAACCTTCTTCTCTCCTGTCAA-3′; R: 5′-GATGTGGATCACGGTGGAC-3′), IRF1 (F: 5′-CTCTGAAGCTACAACAGATGAG-3′; R: 5′-GTAGACTCAGCCCAATATCCC-3′), ISG15 (F: 5′-GAGGCAGCGAACTCATCTTT-3′; R: 5′-AGCATCTTCACCGTCAGGTC-3′), HLA-B (F: 5′-CACTGAGCTTGTGGAGACCA-3′; R: 5′-ATGACCACAACTGCTAGGACA-3′), B2M (F: 5′-CTCGCTCCGTGGCCTTAG-3′; R: 5′-GGATGAAACCCAGACACATAGC-3′), STAT1 (F: 5′-CCATCCTTTGGTACAACATGC-3′; R: 5′-TGCACATGGTGGAGTCAGG-3′), and b-ACTIN (F: 5′-CATGTACGTTGCTATCCAGGC-3′; R: 5′-CTCCTTAATGTCACGCACGAT-3′) genes. Designing IFITM1 and IFITM3 primers to target the unique C-terminal and N-terminal regions, respectively, the specific gene products (including gene name—gene number, transcript, and transcript number) were as follows: IFITM1—ENSG00000185885, IFITM1-202, ENST00000408968.4; IFITM3—ENSG00000142089, IFITM3-201, ENST00000399808.5; IRF1—ENSG00000125347, IRF1-201, ENST00000245414.9; ISG15—ENSG00000187608, ISG15-201, ENST00000379389.4; HLA-B—ENSG00000234745, HLA-B-249, ENST00000412585.7; B2M—ENSG00000166710, B2M-204, ENST00000558401.6; STAT1—ENSG00000115415, STAT1-201, ENST00000361099.7; β-ACTIN—ENSG00000075624, ACTB-201, ENST00000331789.11.

Additional parameters were that all amplicons were expected to have a product size around 150–250 bp and with a melting temperature difference under 1 °C between primers, as multiple genes were run in parallel using the same PCR plates. Designed primer sequences were analyzed by BLAST to ensure there were no additional amplified products. RT-qPCR was run under the same conditions for all the different transcripts. Melting curve analysis was performed to define the specificity of each primer pair (data not shown). Three technical replicates from three biological replicates were set for each sample condition and the raw Ct values of all samples were defined (data not shown). To allow a stringent analysis, Ct expression values were less than one unit difference between each technical triplicate. Relative Ct expression values were normalized with a housekeeping gene, β-ACTIN. The analysis of gene expression was taken from the Ct values obtained by RT-qPCR and normalized into a relative expression using the formula 2^−DCt^, wherein DCt is the difference between the Ct of the transcript of interest and the Ct of b-ACTIN.

### 2.7. Proximity Ligation Assay (PLA)

The wt-SiHa and IFITM1/IFITM3 null cells were grown and processed as described in the immunofluorescence method. Primary antibody pairs from different species were incubated overnight on the fixed, permeabilized, and blocked cells. Depending on requirements of the experiment, the following combination of antibodies were incubated for 18 h at 4 °C: the IFITM1/IFITM3 mouse monoclonal (1:500 dilution) with the SRSF1 rabbit polyclonal (1:250 dilution), or the IFITM1/IFITM3 mouse monoclonal (1:500 dilution) with the rabbit monoclonal anti-BIOTIN (1:200 dilution); the IFITM1/IFITM3 mouse monoclonal (1:500 dilution) with the rabbit polyclonal anti-RPL7a (1:250 dilution). Samples were incubated with the PLA probes; probe-mouse MINUS (DUO92004, Sigma-Aldrich, St. Louis, MI, USA) and probe anti-rabbit PLUS (DUO92002, Sigma-Aldrich, St. Louis, MI, USA) at 37 °C for 1 h. Samples were washed three times with buffer A (150 mm NaCl, 10 mM Tris Base, 0.05% (*v*/*v*) Tween-20, pH 7.4) for 5 min and then samples were incubated with ligation buffer (8 µL of 5× ligation stock (New England Biolabs, Ipswich, MAUSA), 1 µL ligase, and 31 µL of ultrapure water on each coverslip) at 37 °C for 30 min. Samples were again washed three times with buffer A for 5 min and incubated with amplification buffer at 37 °C for 2 h. Reagents were from Duolink in Situ Detection Reagents Green assays (DUO92014, Sigma-Aldrich, St. Louis, MI, USA). Samples were washed twice with buffer A, three times with buffer B (100 mM NaCl, 50 mM Tris base, pH 7.5), and once with 0.01× buffer B. Cells were incubated in DAPI (Invitrogen), diluted at 1:10,000 with 0.01× buffer B for 5 min to stain the nucleus. An additional 2 washes with 0.01× buffer B for 5 min were performed prior to mounting. The fluorescent signal was detected using a Zeiss Axioplan 2 microscope (63× or 100× oil immersion objective). Images were acquired using Micro-Manager 1.4 software. Images were processed in ImageJ 2.0 software.

### 2.8. RNA In Situ Hybridization-PLA (rISH-PLA)

The wt-SiHa cells and IFITM1/IFITM3 null cells were nonstimulated and IFNγ-stimulated for 24 h. Cells were fixed with 4% paraformaldehyde in PBS at RT for 20 min and washed with PBS for 10 min. Then, samples were incubated in 70% (*v*/*v*) ethanol overnight. Cells were then washed in PBS for 30 min and permeabilized with 0.05% CHAPS and 0.4% Triton X-100 for 10 min at RT. Next, samples were treated with hybridization buffer for 30 min at RT. Samples were incubated for hybridization with 40 μL of hybridization buffer (10% (*v*/*v*) formamide, 2X SSC, 0.2 mg/mL *E. coli* 522 tRNAs, 0.2 mg/mL sheared salmon sperm DNA and 2 mg/mL BSA) containing 50 ng of or HLA-B-biotin DNA probe (5′ TGTCCTAGCAGTTGTGGTCATCGGAGCTGTGGTCGCTGCTGTGAT-biotin 3′) (Sigma-Aldrich, St. Louis, MI, USA) in a humidified chamber overnight. Prior to that, 5 μL of probe diluted in water was denatured for 5 min and chilled on ice for 5 min. Then, samples were washed with 2X SSC and 10% formamide at RT, twice with hybridization buffer at 37 °C, 2X SSC at RT, and finally with PBS at RT. Each wash was carried out for 20 min. Next, samples were blocked with 3% BSA and 0.1% saponin in PBS at RT for 30 min. After that, they were incubated with rabbit anti-BIOTIN and mouse anti-IFITM1/IFITM3 at RT for 2 h. The subsequent steps were performed as described in the PLA methodology section (above).

### 2.9. Bioinformatic Analysis of the RNA Sequencing

Three biological replicates of wt-SiHa cells and the isogenic IFITM1/IFITM3 null cells were either left untreated (control) or stimulated for 24 h with 100 ng/mL IFNγ. Total RNA was extracted from frozen cell pellets following the instruction manual (RNeasy Mini kit, Qiagen, Hilden, Germany). Then, RNA from three biological replicates (nonstimulated or IFNγ-stimulated cells) were pooled together in one representative sample and processed by next-generation RNA sequencing of polyA+ enriched RNA.

RNA samples of IFNγ-stimulated and nonstimulated wt-SiHa cells and IFITM1/IFITM3 null cells were processed by Otogenetics, Atlanta, GA, USA for paired end RNA sequencing analysis, using an Illumina HiSeq 2500, and designated 20 million reads. The pair of fastq files obtained from the sequencer were checked for quality control, merged, and processed using CLC Genomic Workbench 12.0 to obtain the total RNA expression levels. GRCh38 was taken as the human reference genome with the following settings: mismatch cost: 2, insertion cost: 3, deletion cost: 3. The results were compared using RNA gene expression for each condition, creating four different scatter plots (Figure 3B) which compared IFNγ-stimulated to nonstimulated conditions, and wt-SiHa cells to IFITM1/IFITM3 null cells. Comparisons were performed taking the transcripts per million (TPM) values as reporting abundances and Log2 (TPM condition 1 versus Log2 (TPM condition 2) as comparison values. An additional heat map was generated with a list of IRDS genes to compare the gene expression of these genes of interest across samples (Appendix A). Following the same parameters, another heat map was created using ggplot2 to compare isoform switches using sum of RNA transcript expression or the RNA transcript expression level of particular transcripts. The color code represented in the heat map shows red when a gene is highly expressed and purple to blue for nonexpressed and underexpressed values. The values in white represent a low level of expression, becoming nonsignificant. Finally, the values in grey are for the genes in which TPM was equal to 0.

### 2.10. Sucrose Gradient Sedimentation

For experiments processing ribosomal fractions using the wt-SiHa cells and IFITM1/IFITM3 null cells the cultures were stimulated with 100 ng/mL IFNγ for 24 h prior to cell harvesting. In the complementation assays (Appendix A)—whereby IFITM1/IFITM3 null were transfected with either the IFITM1 and IFITM3 expression plasmids or empty vector controls—24 h after transfection, the cells were then stimulated with 100 ng/mL IFNγ for 24 h. In all cases, the cells were treated with 50 µg/mL cycloheximide (Merck Chemicals, Darmstadt, Germany) for 30 min. Then, cells were washed in PBS (phosphate buffered saline; 137 mm NaCl, 2.7 mM KCL, 10 mm Na2HPO4, and 1.8 mm KH2PO4) containing 1× RSB, harvested by centrifugation at 7000× *g* rpm for 1 min at 4 °C, and frozen at −80 °C. Lysis was carried out by resuspending the cell pellets in 250 μL of RSB/RNasin buffer and 250 μL PEB (Polysome extraction buffer) buffer. Mechanical disruption of the cell lysate was carried out by passing the lysate though a needle (25 G) five times. Lysates were incubated on ice for 10 min and centrifuged at 10,000× *g* for 10 min at 4 °C. Lysates were processed as described [45]. Sucrose gradients (10–45%) were prepared using a BioComp gradient master. Lysates were applied onto the gradient and centrifuged at 41,000 rpm for 2 h 30 min using a SW41 rotor. Fraction collection was performed using a BioComp gradient station model 153 (BioComp Instruments, Fredericton, NB, Canada). Analysis of ribosomal fractions was performed as biological triplicate. The 10× RSB stock solution contained 200 mM Tris-Hcl (pH 7.5), 1 M KCL, and 100 mM MgCl2. The RSB/RNasin buffer (2 mL) was prepared fresh and contained 200 μL of 10× RSB and 50 μL RNasin (N2511, Promega, Madison, WI, USA). The PEB buffer (10 mL) was made fresh and contained 1 mL of 10× RSB, 50 μL of NP40, and 1 protease inhibitor tablet (Complete Mini-EDTA free EASYpack, 034693159001, Roche/Sigma-Aldrich, St. Louis, MI, USA). In addition, sample preparation for Western blotting required deproteinization of crude samples and was performed using the trichloroacetic acid (TCA) method.

### 2.11. Cloning, Transfection and Affinity Purification of IFITM1 from Heavy Isotope Labeled Cells

IFITM1 cDNA was cloned by PCR into pEXPR-IBA105 expression vector containing an SBP tag at the N-terminus of the coding region (SBP vector, IBA Lifesciences, Göttingen, Germany). Cells were grown as biological triplicates for 10 days with 5 passages in RPMI-stable isotope labeling with amino acids in cell culture (SILAC) media before transfection (Dundee Cell Products, Dundee, UK). Cells were isotopically labeled with light media (l-[12C614N4] arginine (R0) and l-[12C614N2] lysine (K0)) and heavy media (l-[13C614N4] arginine (R6) and l-[13C614N2] lysine (K6)). For transfection, cells were grown to approximately 80% confluency in light and heavy media and transfected using Attractene (#301007, Qiagen, Hilden, Germany) with SBP-empty vector (control cells) and SBP-IFITM1 (Appendix A). At 24 and 48 h after transfection, cells were washed twice in ice cold PBS and scraped into 0.1% Triton buffer for 30 min on ice. Equal amounts of protein were used for performing the pull down. Total protein extracts were measured by Bradford assay. For affinity purification, the cells were washed twice in cold PBS and scraped directly into IP buffer (100 mM KCl, 20 mM HEPES pH 7.5, 1 mM EDTA, 1 mM EGTA, 0.5 mM Na3VO4, 10 mM NaF, 10% (*v*/*v*) glycerol, protease inhibitor mix, and 0.1% Triton X-100). The lysate was then incubated for 30 min on ice and centrifuged at 13,000 rpm for 15 min at 4 °C. Then, cell lysate was added to Streptavidin agarose conjugated beads (Millipore, MA, USA) and incubated for 2 h with gentle rotation. The proteins were eluted from the beads using elution buffer (20 mM HEPES pH 8, 2 mM DTT, and 8 M Urea). The eluted samples (whole volume) from light and heavy SILAC medium-labeled cells were mixed together.

### 2.12. Peptide Generation Using FASP

Proteins eluted after SBP pull-down were processed using the filter-aided sample preparation protocol method (FASP) [46] according to a workflow described in Gómez-Herranz et al. [43]. Briefly, protein concentration was determined using the RC-DC protein assay (Bio-rad, California, CA, USA). Approximately 100 µg of protein dissolved in 20 mM HEPES pH 8, 2 mM DTT, and 8 M Urea was added to a 10 kDa spin filter column (Microcon, Merck-Millipore, Burlington, MA, USA), on-filter reduced, alkylated, and trypsin-digested to peptides. Tryptic peptides were then desalted using C18 microspin columns (Harvard Apparatus, Cambridge, MA, USA) [43].

### 2.13. LC-MS/MS Analysis of SILAC Labeled Samples

Tryptic peptides from isotopically labeled cells were separated using an UltiMate 3000 RSLCnano chromatograph (Thermo Fisher Scientific, Waltham, MA, USA). Tryptic peptides were loaded onto a precolumn (μ-precolumn, 30 µm i.d., 5 mm length, C18 PepMap 100, 5 µm particle size, 100 Å pore size) and separated using an Acclaim PepMap RSLC column (75 µm i.d., length 500 mm, C18, particle size 2 µm, pore size 100 Å). Tryptic peptides were separated by a linear gradient of mobile phase B (B = 80% (*v*/*v*) acetonitrile (ACN), 0.08% (*v*/*v*) formic acid (FA) in water) and A (A = 0.1% (*v*/*v*) FA in water) as follows: 2% B over 4 min, 2–40% B over 64 min, and 40–98% B over 2 min. The flow rate was 300 nL/min. Tryptic peptides eluting from the column were injected into an Orbitrap Elite (Thermo Fisher Scientific, Waltham, MA, USA) operating in Top20 data-dependent acquisition mode. Scanning was set to 400–2000 *m*/*z*, performed at a 120,000 resolution. The AGC target was 1 × 10^6^ with a 200 ms injection time and twenty data-dependent MS2 scans (1 microscan, 10 ms injection time and 10,000 AGC).

### 2.14. Database Searching and Analysis

The SILAC data were processed using Proteome Discoverer 1.4 (Thermo Fisher Scientific, Waltham, MA, USA) and the Mascot search engine with the following search settings: a human database, Swiss-Prot (April 2016); enzyme-trypsin; 2 missed cleavage sites; a precursor mass tolerance of 10 ppm; a fragment mass tolerance of 0.6 Da; modification included: carbamidomethyl [C], oxidation [M], acetyl [protein N-terminus]. The search results were used to generate the final report with a 1% FDR on both PSM and peptide groups. SILAC labels of R6 and K6 were chosen for heavy, and R0 and K0 for light. The relative quantification value was represented as heavy/light ratio (Appendix A).

## 3. Results

### 3.1. Defining New Protein–Protein Association with IFITM Proteins

IFITM1/2/3 are IFN-stimulated molecules involved in human cancer development, including cervical carcinogenesis [47]. In addition, gene expression of cervical cancer specimens reveals an inverse correlation between IFNγ and lymph node metastases [48]. For this study, a cell model that was engineered in our previous publication is used to investigate the role of IFITMs—SiHa cells originating from squamous cell carcinoma of the cervix which express IFITM1 and IFITM3 at detectable endogenous levels with a higher expression upon IFNγ stimulation (Figure 1A). As IFITM2 protein was detected at neither the endogenous level nor after IFNγ stimulation in the parental wt-SiHa cells (Appendix A), we use the genetically engineered IFITM1 and IFITM3 knockout cell line (hereafter referred to as IFITM1/IFITM3 null) by using CRISPR/Cas9 technology.

The immune-related IFITM1/2/3 proteins have conserved amino acid sequences and form homo- and hetero-oligomers [49,50,51], suggesting that they may cooperate together. Thus, to rule out any possible functional redundancy of IFITM1 and IFITM3, we used the IFITM1/IFITM3 knockout cells. A representative IFITM1/IFITM3 null clone showed that IFITM1/3 protein expression diminished under the level of detection by Western blot (Appendix A) and immunofluorescence (Appendix A). More details on the design and screening of this cell model can be found in our previous publication [43].

IFITM1 is the main IFITM family member reported in tumor progression and poor prognosis for many human cancer types [18,29,34,35,36,40,52]. Because IFITM interactome is not well defined, here, we asked whether new binding partners can emerge by employing SILAC coupled with MS. The full-length IFITM1 gene was cloned into an SBP-tagged expression vector [53] to allow for its affinity capture from crude lysates after transfection (Appendix A). The affinity-purification protocol using streptavidin beads was designed to capture SBP-IFITM1 and associated proteins. Cells were isotopically labeled using SILAC RPMI media containing 13C-labeled arginine and lysine amino acids (R6K6), while the SBP empty vector that was used as a negative control was expressed in cells containing unlabeled amino acids.

Under these conditions, SBP-IFITM1 protein was detected in the affinity capture from cells grown in heavy media at 24 h and 48 h after transfection. In addition, common binding proteins were identified (Appendix A) and, more importantly, SBP-IFITM1 peptides were dominantly identified after MS analysis (Appendix A). This internal control highlights that the methodology enriches and identifies the IFITM1 bait protein.

We focused on the common interacting proteins detected in the SBP-IFITM1 purification between these two time points. The most common targets were the splicing regulatory factors within the SRSF superfamily of serine–arginine-rich splicing factors: SRSF1, SRSF2, SRSF3, SRSF6, and U2AF1 (Appendix A; tryptic peptides are highlighted). SRSF1 is a nuclear splicing factor that can shuttle to the cytoplasm upon binding to certain target transcripts to facilitate protein translation [54,55]. Interestingly, one of the tryptic peptides derived from the SRSF1 isoform ASF-1, located in the C-terminus, is characteristic of the cytosolic form [56,57] (Appendix A; SRSF1 tryptic peptide is underlined in red). Although the SRSF1:IFITM1/3 signal is predominantly in the cytosol, we identified other nuclear protein candidates in our pull down, suggesting a possible nuclear location of IFITM1/3. Nonetheless, we did not further investigate whether IFITM1/3 proteins enter into the nucleus.

SRSF1 is the archetype component of the SR family of splicing factors [58]. Thus, the endogenous IFITM1 and SRSF1 coassociation was further validated by PLA, which can detect two endogenous protein–protein interactions in situ [59], not relying on the expression of exogenous vector constructs. In this case, IFITM1 and IFITM3 were simultaneously detected to capture all of the IFITM endogenous interactions with SRSF1. A basal level of IFITM1/IFITM3-SRSF1 foci can be observed in nonstimulated wt-SiHa cells (Figure 1C vs. Figure 1A or Figure 1B; quantified in Figure 1I). The stimulation with IFNγ for 24 h elevated the number of foci that predominate in the cytoplasm (Figure 1C vs. Figure 1D; quantified in Figure 1I). The cytoplasmic localization of the majority of IFITM1/IFITM3:SRSF1 foci is consistent with the tryptic peptides for the cytoplasmic isoform SRSF1 detected in the affinity capture (Appendix A; SRSF1 and Appendix A).

To indicate that the method is not detecting artefactual signals, the IFITM1/IFITM3 null cells did not exhibit IFITM1/IFITM3-SRSF1 foci, not even after IFNγ stimulation (Figure 1E–H; quantified in J). Furthermore, SRSF1 protein itself is not induced by IFNγ treatment (Figure 1K), nor is SRSF1 depleted in the IFITM1/IFITM3 null cells (Figure 1K). These data indicate that the IFITM1/3 proteins are predominantly responsible for the IFNγ dependent induction of the SRSF1-IFITM1/IFITM3 foci (Figure 1D). The results suggest that the SBP-IFITM1 pull down (Appendix A) captures authentically colocalizing IFITM1:SRSF1 complexes.

### 3.2. Association of IFITM1/3 Proteins with HLA-B RNA

In our previous report, we described that endogenous HLA-B was deficiently expressed in IFNγ-stimulated cells lacking IFITM1/3 expression [43]. In addition, the proposed SRSF1 interacting protein (Appendix A) was involved in splicing, RNA binding, and transport. Then, we asked whether IFITM1/3 was associated with mRNA in vivo. For this purpose, we examined the interaction between IFITM1/3 with HLA-B RNA as a potential mechanism to account for IFITM1/3-dependent HLA-B protein synthesis in response to IFNγ. Protein–RNA proximity ligations (rISH-PLA) [60] were used to measure IFITM1 protein–HLA-B RNA interactions. The IFITMs and HLA-B are ISG genes, but the IFNγ-induced expression of HLA-B was impaired in the absence of IFITM1/3 expression [43]. Taking this into account, wt-SiHa cells and IFITM1/IFITM3 null cells were nonstimulated and IFNγ-stimulated for 24 h (Figure 2). Cells were processed as indicated in the methods and incubated with antibodies to IFITM1/3 and biotinylated probe designed to bind specifically the HLA-B mRNA.

IFITM1/3:HLA-B rISH-PLA images identify the protein–RNA association foci (green) and DAPI was used for nuclear staining (blue). Cells were incubated as negative controls (Figure 2A–C,F–H) without the RNA probe (Figure 2A,F), using primary antibodies only (Figure 2B,G), or using secondary antibodies only (Figure 2C,H). Cells were incubated with both IFITM1/IFITM3 and biotin antibodies to define protein–RNA foci in nonstimulated cells (Figure 2D,I) or in IFNγ-stimulated cells (Figure 2E,J). Endogenous interactions between HLA-B mRNA and IFITM1/3 proteins were shown in wt-SiHa cells (Figure 2A–E) and barely any interaction was observed in IFITM1/IFITM3 null cells (Figure 2F–J). Representative quantification was observed of the protein–RNA interaction foci per cell in presence or absence of IFNγ stimulation in wt-SiHa cells (Figure 2K) and IFITM1/IFITM3 null cells (Figure 2L). Compiling both results (Figure 1 and Figure 2), the data suggest that IFNγ can induce the association of IFITM1/3 proteins with SRSF1 protein and HLA-B RNA. In addition, HLA-B expression is attenuated in absence of IFITM1/IFITM3 expression (Figure 2M). These results start explaining why HLA-B protein synthesis is attenuated in the IFITM1/IFITM3 cells [43].

### 3.3. Exploring RNA Transcripts in wt-SiHA and IFITM1/IFITM3 Null Cells

Next, we asked whether attenuated HLA-B protein synthesis in the IFNγ-stimulated IFITM1/IFITM3 null cells [43] was related to reduced mRNA levels. This suggests that the dominant mechanism for IFITM1/3 protein signaling was a transcription-dependent mechanism. To address this question, a targeted, quantitative RT-PCR assay, with biological triplicate samples from wt-SiHa and IFITM1/IFITM3 null cells nonstimulated or IFNγ-stimulated, was measured, each sample in technical triplicate. In addition to the HLA-B transcript, we also included additional IFNγ-stimulated genes as controls: STAT1, B2M, and IRF1. Further, we validated the IFITM1/3 knockout cells, IFITM1 and IFITM3. Of note, STAT1, B2M, IFITM1, and HLA-B are IRDS genes. 

The positive controls highlight equivalent mRNA levels observed upon IFNγ induction of STAT1 (master signal transducer of the IFNγ pathway) and B2M (MHC class I subunit) in both wt-SiHa and IFITM1/IFITM3 null cells (Figure 3A(I,II)). This is consistent with our previous data, wherein we observed IFITM1/IFITM3-independent synthesis of STAT1 and B2M proteins using pulse-SILAC mass spectrometry [43]. In addition, we found that IRF1 (transcriptional activator of the secondary ISG response in the IFNγ pathway) protein synthesis was partially attenuated in IFNγ-stimulated IFITM1/IFITM3 null cells using pulse-SILAC mass spectrometry [43]. However, its mRNA levels are equivalently induced by IFNγ in both cell lines (Figure 3A(V)). Moreover, there is a discreate increase in IFITM1 and IFITM3 mRNA abundance in IFITM1/IFITM3 null cells, nonstimulated vs. after IFNγ stimulation (Figure 3A(III,IV), respectively). Although gene edits in IFITM1 and IFITM3 disrupt protein expression in IFITM1/IFITM3 null cells (Figure 2M), aberrant mRNA may be produced after IFNγ treatment. These data might be expected if gene editing creates indels that result in stop codon mutations or alterations in the amino acid sequence which cause mRNA degradation by NMD [61].

Although HLA-B protein synthesis was not detected after IFNγ stimulation in IFITM1/IFITM3 null cells using pulse-SILAC mass spectrometry [43], there was an equivalent 100–250-fold increase in the relative levels of HLA-B mRNA in both the wt-SiHa cells and IFITM1/IFITM3 null cells (Figure 3A(VI)). This later result suggests that the mechanism driving the reduced synthesis of HLA-B proteins after IFNγ stimulation in IFITM1/IFITM3 null cells is independent of the transcript products. Because HLA-B mRNA are similarly induced by IFNγ in the IFITM1/IFITM3 null cells compared to wt-SiHA cells (Figure 3A(VI)), we focused next on understanding what might be linked to defects in protein synthesis in the IFITM1/IFITM3 null cells.

After these preliminary results, using the same RNA preparations used for the RT-PCR analysis (Figure 3A), we performed a global gene expression analysis using RNA-seq (Figure 3B) to investigate whether a lack of IFITM1/3 expression, overall, changes other mRNA transcript products.

The wt-SiHa cells exhibited the expected induction of standard IFNγ-inducible genes, including HLA-B, IFITM1, STAT1, and IRF1 (Figure 3B; gene names highlighted in red). Loss of IFITM1/3 protein expression does not decrease these transcripts (Figure 3B(IV); highlighted in red). In addition to these classic IFNγ-inducible genes, other highly induced transcripts after IFNγ treatment are also revealed to be IFITM1/IFITM3-independent. These include: HLA-DRB1, HLA-DRA, CXCL10, and GBP5 (Figure 3(I,IV); highlighted in black). These genes are related to tumor immunity or pathogen restriction [62,63,64]. Nonetheless, globally, no major differences were observed in the total amount of transcripts in the cells lacking IFITM1/3 expression compared to wt-SiHa cells (Figure 3B).

By comparing the mRNA ratio in wt-SiHa cells to IFITM1/IFITM3 null cells, the dominant suppressed gene detected was IFITM1 (Figure 3B(II,III); highlighted in red). This is consistent with IFITM1 being a dominant interferon-inducible target and IFITM1 gene mutation through guide RNA editing (as described in [43]) reduces the basal levels of the IFITM1 mRNA.

Furthermore, we focused on the mRNA abundance of HLA-B and other IRDS genes for which expression seems to be mediated by IFITM1/3. These genes are derived from Appendix A, and they are a subset of IFN-responsive genes linked to chemoradiation resistance that include IFITM1 (but not IFITM3) and HLA-B [65]. In this regard, a heat map quantifying the expression of selected IRDS genes are highlighted in Appendix A. Overall, there is a substantial induction of IRDS genes after IFNγ stimulation in both the wt-SiHa cells and the IFITM1/IFITM3 null cells (Appendix A).

### 3.4. Reduction in 80S Ribosomal RNA Levels in the IFITM1/IFITM3 Null Cells Using Sucrose Gradient Sedimentation

We aimed to investigate whether IFITM1/3 proteins can modulate HLA-B expression by other mechanisms not involving global transcript level changes. We have previously shown that HLA-B interacts with IFITM1/3 [43] and, in this report, we have identified a new interacting (or in close proximity) partner for IFITM1, SRSF1, which, interestingly, mediates splicing and regulates translation. Taken all together, we investigated whether IFITM1/3 proteins may modulate HLA-B expression through intervention in the translational process.

In the attempt to study the relation of IFITM1/3 with HLA-B protein translation, we first asked whether IFITM1/3 proteins were present in the ribosome by studying their potential interaction with RPL7a protein, a ribosome-specific protein used as a subcellular marker. RPL7a is a component of the 60S ribosomal [66] which associates with the endoplasmic reticulum [67]. By performing a PLA, there was a significantly increased protein–protein association of IFITM1/IFITM3:RPL7a after IFNγ-stimulation in the cytosol compared to nonstimulated wt-SiHa cells (Figure 4C,D,I). In contrast, there was almost no detectable signal in the IFNγ-stimulated IFITM1/IFITM3 null cells (Figure 4H,J). It is worth mentioning that IFITM1/IFITM3 proteins accumulate perinuclearly upon IFNγ stimulation (Appendix A, arrowhead in merge composition with IFNγ-stimulated in wt-SiHa cells), which is consistent with a possible ribosomal localization after the activation of the IFNγ signaling.

Given that IFITM1 can regulate HIV mRNA translation [42], we set out to determine whether global defects in ribosomal integrity was linked to defects in HLA-B protein synthesis [43] in the IFITM1/IFITM3 null cells. An analysis of the ribosomal profile from the wt-SiHa and the IFITM1/IFITM3 null cells was first used to define ribosomal integrity. Then, cell lysates from IFNγ-stimulated cells were subjected to sucrose gradient sedimentation under conditions that maintain ribosome integrity (Figure 5). Immunoblots of the polysomal fractions indicated that IFITM1/3 proteins can be detected in the 40S, 60S, and 80S fractions from wt-SiHa cells but not from the IFITM1/IFITM3 null cells (Figure 5B). 

The representative A254 nm peaks of 40S, 60S, 80S and polysome profile are depicted in Figure 5A. Interestingly, the A254 absorbance scan of the 80S subunit fraction in the IFITM1/IFITM3 null cells was reproducibly lower than in wt-SiHa cells in three independent biological replicates (Figure 5A,C,D; quantified in Figure 5E and Appendix A). Remarkably, transient transfection of IFITM1 and IFITM3 genes for 24 h into the IFITM1/IFITM3 null cells, in three independent biological replicates, can partially restore 80S ribosomal RNA levels as defined by increase in A254 nm (Appendix A). These data suggest a selective impairment in the RNA component of the 80S ribosome in the absence of IFITM1/3 expression. However, the defects observed in 80S integrity are not an irreversible feature of the IFITM1/IFITM3 null cell line.

## 4. Discussion

Several studies have postulated that high expression of the IFN-stimulated IFITM1 and IFITM3 proteins can stimulate more aggressive growth of human cancers [40,68,69,70]. In addition, IFITM1 is upregulated during the development of radiation resistance [18]. By contrast, IFITM1 activity is linked to growth suppression in cervical cancers [71,72]. In this regard, in our previous work, we identified a distinct subgroup of cervical cancer patients wherein IFITM1/IFITM3 proteins were highly expressed in cervical cancers. However, their expression was inversely correlated to the number of metastatic lymph nodes [43]. With this premise, we aimed to understand why IFITM1/IFITM3 double negative patients are also susceptible to developing aggressive tumors.

To broaden the limited molecular understanding of IFITM1 and IFITM3 function(s), in our previous report, we developed an MS assay to determine whether IFITM1 and IFITM3 play a role in the response of cells to IFN-induced protein synthesis [43]. As a result, we identified HLA-B, an MHC class I molecule, as one of the few dominant downstream effectors of IFITM1/3 [43]. Moreover, a decreased expression of HLA-B on the cell surface in cells with a complete loss of IFITM1/3 expression was observed. Taken all together, the data suggested that the loss of IFITM1/3 expression would attenuate antigen presentation in vivo [43]. However, in this current study, we are interested in which cellular process explains why HLA-B exhibits lowered synthesis in the IFITM1/IFITM3 null cells. With this, we could further comprehend other mechanisms by which the antigen presentation pathway is regulated.

To begin with, we start tackling this question by identifying new interacting partners for IFITMs. Employing an affinity capture assay coupled to pulse-SILAC mass spectrometry, we identified multiple nuclear proteins, such as members of the SR family of splicing factors (SRSF1, SRSF2, SRSF3, SRSF6, and U2AF1), as novel candidates that associate with IFITM1 proteins or form a complex (Appendix A). A proximity ligation assay confirmed the association between IFITM1/3 and SRSF1 (Figure 1). However, immunoblot did not show major difference in the expression of SRSF1 in cells lacking IFITM1/3 expression (Figure 1K). Interestingly, in addition to be implicated with the splicing machinery, the splicing factors identified are also implicated in the mRNA transport [73]. Although SRSF2 is not considered as nuclear to cytoplasmic shuttled proteins, it has a C-terminal enriched in arginine and serine; these amino acid motifs are crucial for the nuclear exit of the mentioned splicing factors (Appendix A; SRSF2). Thus, we studied the possible implication of IFITM1/3 proteins regulating the transcript levels. To address this possibility, the performed RNA expression analysis did not reveal a dominating defect in either the HLA-B gene expression or in the global transcript production by comparing IFITM1/IFITM3 null cells with wt-SiHa cells (Figure 3). These data suggested that defects in HLA-B protein synthesis in cells lacking IFITM1/3 expression are not selectively due to the suppression of IFN-stimulated gene induction, even though IFITM1/3 associate with the HLA-B mRNA (Figure 2). IFITMs have not reported an RNA binding domain; thus, we hypothesize that they may associate with SRSF1, which contains two RRMs, to form an IFITM1/3:SRSF1:HLA-B (protein–protein-RNA) complex.

SRSF factors are also implicated in guiding the mRNA products to the ribosome [77] and a ribosomal localization of IFITM1/3 proteins is also inferred due to the coassociation of RPL7a and IFITM1/3 proteins using proximity ligation assays (Figure 4). RPL7a is a component of the 60S ribosomal subunit that contains nucleic acid-binding domains, and it is implicated in regulating the expression of mRNAs [74,75].

Next, we interrogated whether IFITM1/3 could be selectively mediating the protein synthesis (which is more representative for HLA-B) by modulating their translation. To find some answers, we evaluated the ribosomal profile using ultracentrifugation sedimentation of cell lysates from IFNγ-stimulated wt-SiHa cells and IFITM1/IFITM3 null cells to isolate ribosomal constituents. Our results suggest a role for IFITM1/IFITM3 in regulating the integrity of 80S ribosomal subunits, partially restored with the exogenous transient expression of IFITM1/3 (Figure 5 and Appendix A). During the course of our studies, it was shown that IFITM protein expression reduces HIV-1 viral protein synthesis by preferentially excluding viral mRNA transcripts from translation [42]. This study supports our vision wherein an exciting new role for IFITM family members operating at the level of protein translation and/or mRNA transportation is emerging. Although we focused on the role of IFITM1/3 in regulating HLA-B, there may be other cancer- and virus-originated proteins for which expression may be mediated by IFITM1/3.

IFITM1/3 could regulate HLA-B protein levels by affecting several processes, such as alternative splicing, capping, mRNA export, intervention of small regulatory RNAs, or protein synthesis. Here, we are starting to reveal an alternative novel role for IFITM1/3 proteins that may help in understanding the complicated mechanism by which MHC class I molecules are regulated. Although it is striking how suppression of only two proteins (IFITM1 and IFITM3) can impact the 80S fraction, the biological reasoning behind this is complicated to convey, as many factors may be implicated. Based on our investigation, we hypothesize that IFITM1/IFITM3 may affect the remodeling of the ribosome through selectively transporting mRNA products. Future research should investigate further how IFITM1/3 mediate the HLA-B expression; for instance, it should be addressed whether IFITM1/3 are required to form the protein complexes guiding mRNA, whether IFITM1/3 guides the mRNA to the ribosome, or whether IFITM1/3 exclude the mRNA from loading into the ribosome.

However, our findings emphasize the importance of IFITM1/IFITM3 in mediating HLA-B protein expression through interactions with its mRNA. Such RNA-dependent effects on HLA-B synthesis might impact the global number of neo-antigens presented to CD8+ T cells, otherwise promoting immune scape. From a cervical cancer perspective, MHC class I depletion is not correlated with HPV positive tissues [76,77]; it is suggested that these events originate at different stages of cervical cancer progression. Independent studies have identified an impaired MHC class I expression in cervical tumors (and other cancer types); particularly, loss of MHC class I molecules occurs in metastatic lesions compared to primary tumors [78,79], including chromosomal aberrations [80]. This is of particular relevance for our study, as it supports the hypothesis whereby IFITM1/3 are able to regulate, at least partially, the expression of MHC class I proteins. Remarkably, some studies have revealed that HLA expresses deficiencies in lymph node metastases compared to primary tumors [81,82], which is consistent with the concept that a lack of IFITM1/IFITM3 would lead to diminished HLA expression [43].

## Figures and Tables

**Figure 1 biomolecules-12-01090-f001:**
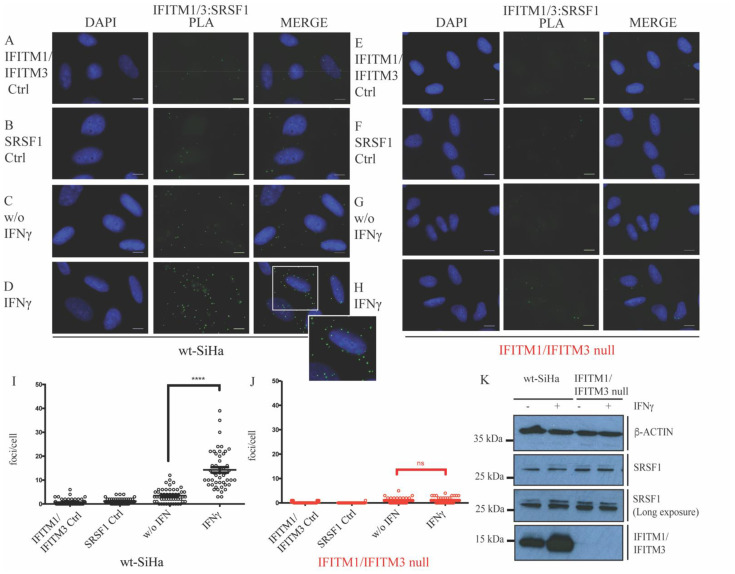
Evaluation of the IFITM1/IFITM3:SRSF1 protein–protein expression and interaction in situ after stimulation with IFNγ. Proximity ligation assays were used to study the endogenous interaction between SRSF1 and IFITM1/3 proteins in wt-SiHa (**A**–**D**) and IFITM1/IFITM3 null cells (**E–H**). IFITM1/3:SRSF1 PLA images identify the protein–protein association foci (depicted in green) and DAPI was used for nuclear staining (depicted in blue). (**A**,**B**,**E**,**F**) Cells were incubated as negative controls using IFITM1/IFITM3 or SRSF1 antibodies only. (**C**,**G**) Cells were incubated with both IFITM1/IFITM3 and SRSF1 antibodies to define protein–protein foci in nonstimulated cells. (**D**,**H**) Cells were incubated with both IFITM1/IFITM3 and SRSF1 antibodies to define protein–protein foci in IFNγ-stimulated cells. Magnification of selected area at the bottom right of merge (**D**). Representative quantification of the protein–protein interaction foci per cell in presence or absence of IFNγ stimulation in wt-SiHa cells (**I**) and IFITM1/IFITM3 null cells (**J**). At least 50 cells were counted in each condition. Statistical study was performed with one-way ANOVA and Bonferroni correction (**** *p* < 0.0001; ns, not significant). *n* = 3. Scale bar: 10 μm. (**K**) Immunoblots examining the endogenous levels of SRSF1 protein in nonstimulated or IFNγ-stimulated wt-SiHa cells or IFITM1/IFITM3 null cells.

**Figure 2 biomolecules-12-01090-f002:**
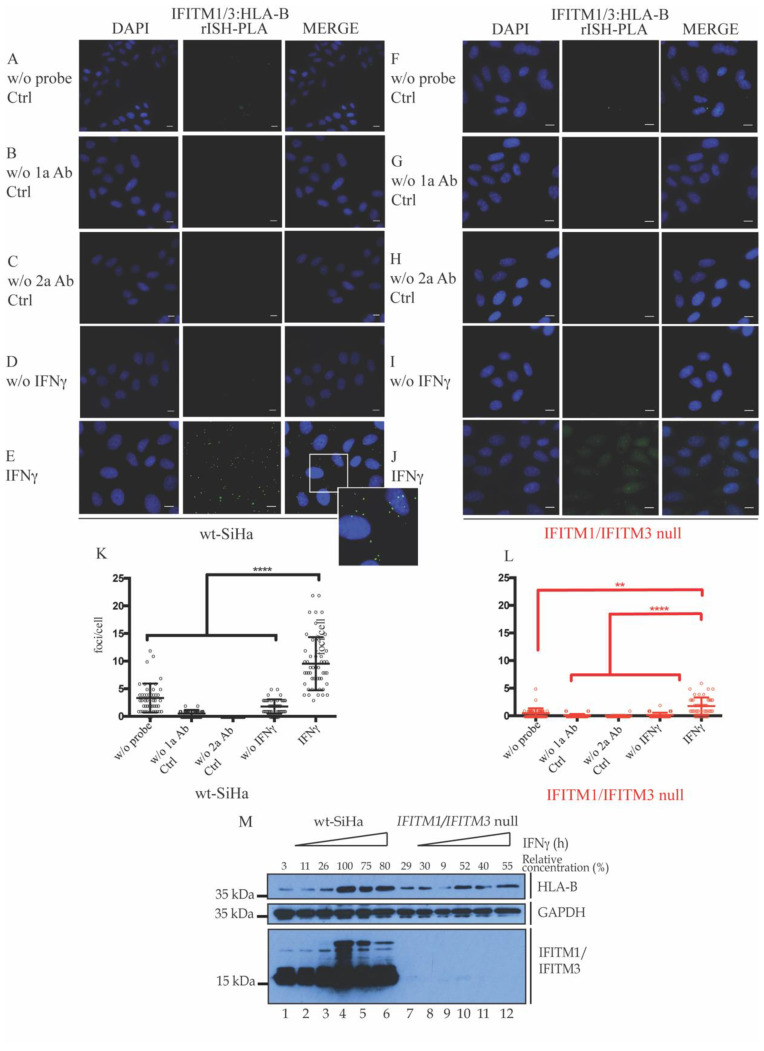
Evaluation of the IFITM1/IFITM3:HLA-B protein–RNA interaction in situ after stimulation with IFNγ by rISH–PLA. RNA in situ hybridization—PLA assays were used to study the endogenous interaction between HLA-B mRNA and IFITM1/3 proteins in wt-SiHa cells (**A**–**E**) and IFITM1/IFITM3 null cells (**F**–**J**). A biotinylated probe was designed to bind specifically the HLA-B mRNA. IFITM1/3:HLA-B rISH-PLA images identify the protein–RNA association foci (green) and DAPI was used for nuclear staining (blue). (**A**–**C**,**F**–**H**) Cells were incubated as negative controls without the RNA probe (**A**,**F**), using primary antibodies only (**B**,**G**), or secondary antibodies only (**C**,**H**). Cells were incubated with both IFITM1/IFITM3 and biotin antibodies to define protein–RNA foci in nonstimulated cells (**D**,**I**) or IFNγ-stimulated cells (**E**,**J**). Magnification of selected area at the bottom right of merge (**E**). Representative quantification of the protein–RNA interaction foci per cell in presence or absence of IFNγ stimulation in wt-SiHa cells (**K**) and IFITM1/IFITM3 null cells (**L**). At least 50 cells were counted in each condition. Statistical study was performed with one-way ANOVA and Bonferroni correction (**** *p* < 0.0001, ** *p* < 0.01). *n* = 3. Scale bar: 10 μm. A time course of 0, 4, 6, 24, 48, and 72 h with 100 ng/mL INFγ stimulation in wt-SiHa and IFITM1/IFITM3-null cells was conducted to measure levels of HLA-B and IFITM1/IFITM3. GAPDH was used as loading control. Relative concentration (%) was measured using Fiji software by taking the higher IFITM1/IFITM3 intensity (lane 4) as 100% intensity, and other lanes were normalized to lane 4 (indicated on top of HLA-B panel) (**M**).

**Figure 3 biomolecules-12-01090-f003:**
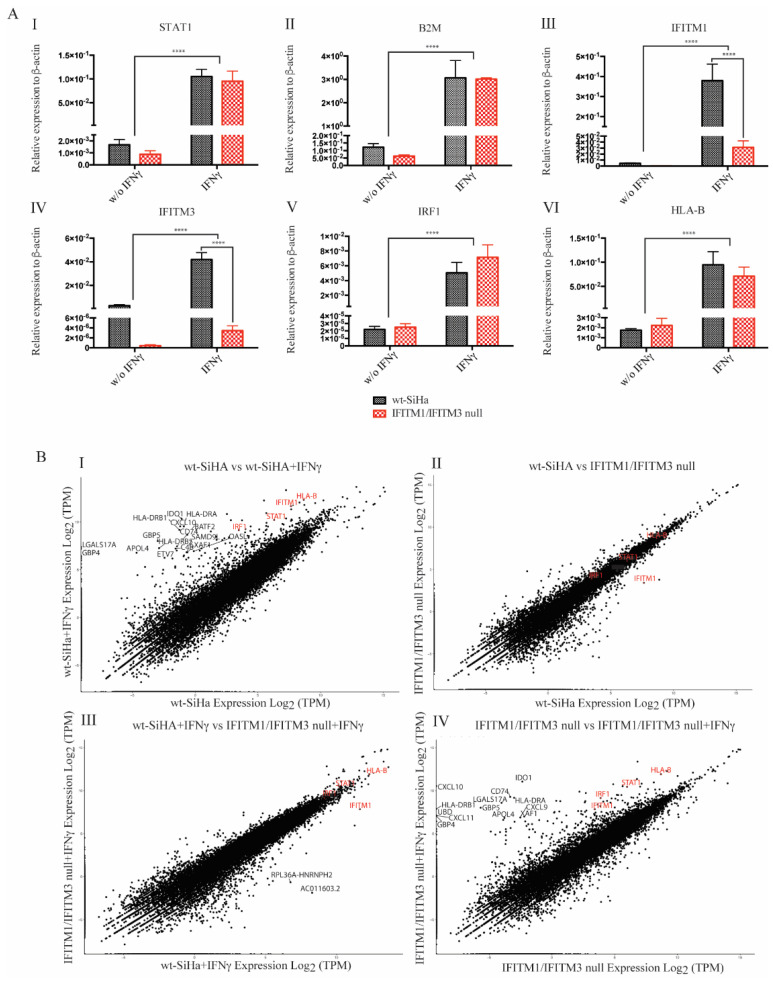
Validation of the transcript levels of IFNγ-stimulated genes in wt-SiHa cells and IFITM1/IFITM3 null cells, unstimulated or IFNγ-stimulated, for 24 h (**A**). STAT1 (**I**), B2M(**II**), IFITM1 (**III**), IFITM3 (**IV**), IRF1 (**V**), and HLA-B (**VI**) mRNA was extracted and reverse transcribed into cDNA for quantification by RT-qPCR in wt-SiHa (in black) and IFITM1/IFITM3 null cells (in red). Measurements were performed in untreated cells and after stimulation with 100 ng/mL IFNγ for 24 h. Error bars are a representation of the variability between three biological replicates. Each biological sample was run in three technical replicates. β-ACTIN was used to normalize the mRNA expression between samples. Statistical study was performed with two-way ANOVA (**** *p* < 0.0001). (**B**) Comparison of total transcript count in wt-SiHa cells and IFITM1/IFITM3 null cells. Fastq files were imported into CLCBio Genomics workbench 12.0. The sequencing reads were used as the input file; all transcript reads detected were taken to generate the final transcript count for each gene (Appendix A). Comparisons of all transcripts identified are plotted for the following conditions: (**I**) nontreated wt-SiHa cells vs. IFNγ-stimulated wt-SiHa cells, (**II**) nontreated wt-SiHa cells vs. nontreated IFITM1/IFITM3 null cells, (**III**) IFNγ-stimulated wt-SiHa cells vs. IFNγ-stimulated IFITM1/IFITM3 null cells, and (**IV**) nontreated IFITM1/IFITM3 null cells vs. IFNγ-stimulated IFITM1/IFITM3 null cells. Each plot has STAT1, IFITM1, IRF1, ISG15, and HLA-B transcripts highlighted in red. Other significantly induced transcripts are additionally highlighted in black. Transcript expression is measured in Log2 of transcripts per million (TPM).

**Figure 4 biomolecules-12-01090-f004:**
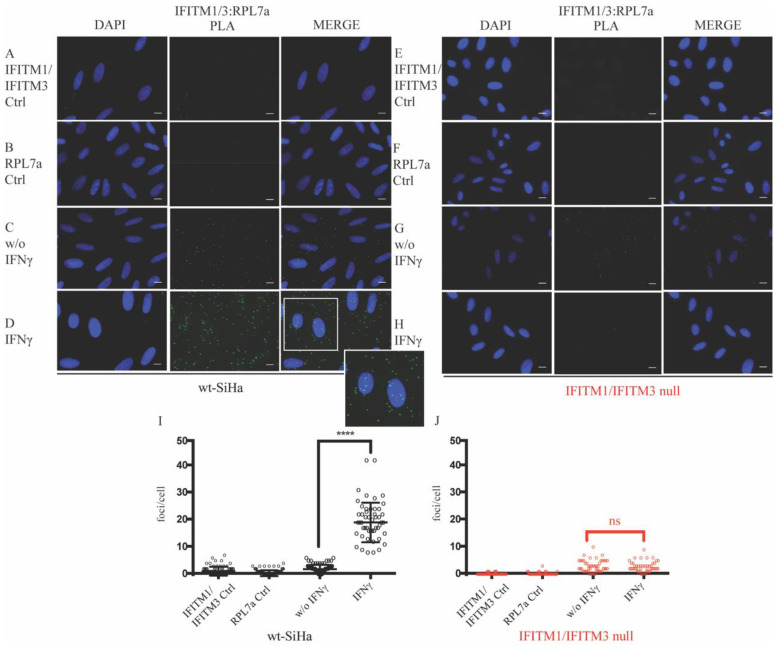
Evaluation of the IFITM1/IFITM3:RPL7a protein–protein expression and interaction in situ after stimulation with IFNγ. Proximity ligation assays were used to study the endogenous interaction between RPL7a and IFITM1/IFITM3 proteins in wt-SiHa (**A**–**D**) and IFITM1/IFITM3 null cells (**E**–**H**). IFITM1/3:RPL7a PLA images identify the protein–protein association foci (green) and DAPI was used for nuclear staining (blue). (**A**,**B** and **E**,**F**) Cells were incubated as negative controls using IFITM1/IFITM3 or RPL7a antibodies only. (**C**,**G**) Cells were incubated with both IFITM1/IFITM3 and RPL7a antibodies to define protein–protein foci in nonstimulated cells. (**D**,**H**) Cells were incubated with both IFITM1 and RPL7a antibodies to define protein–protein foci in IFN-stimulated cells. Magnification of selected area at the bottom right of merge (**D**). Representative quantification of the protein–protein interaction foci per cell in presence or absence of IFN stimulation in wt-SiHa (**I**) and IFITM1/IFITM3 null (**J**). At least 50 cells were counted for each condition. Statistical study was performed with one-way ANOVA and Bonferroni correction (**** *p* < 0.0001; ns, not significant). *n* = 3. Scale bar: 10 μm.

**Figure 5 biomolecules-12-01090-f005:**
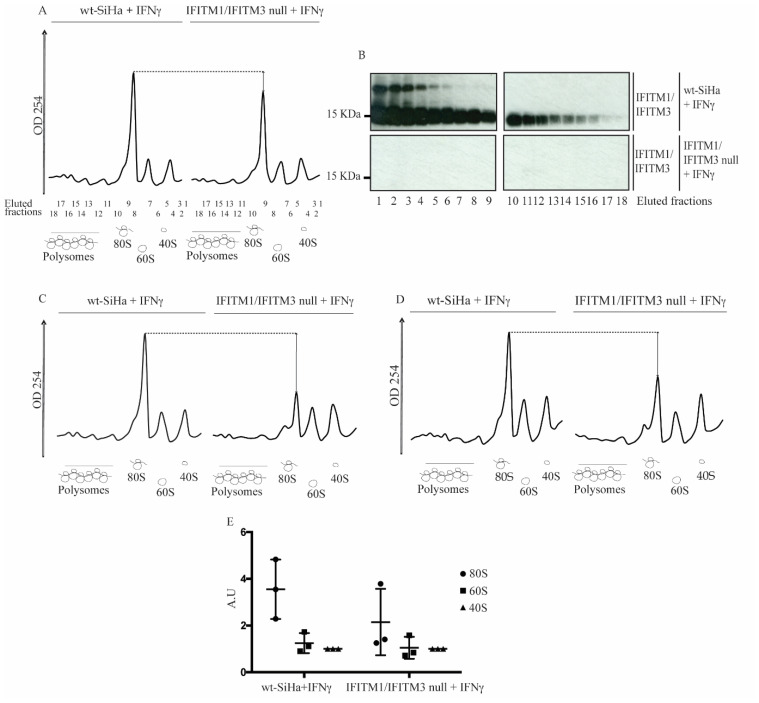
Ribosomal integrity analysis in IFNγ-stimulated wt-SiHa cells and IFITM1/IFITM3 null cells. (**A**,**C**,**D**). The wt-SiHa cells and IFITM1/IFITM3 null cells were stimulated in three biological replicates with 100 ng/mL IFNγ for 24 h. The cells were lysed with ribosome stabilization buffer and applied to a 10–45% sucrose gradient to separate individual components of the large and small ribosomal subunits and polysomal structures. Eluted fractions (numbered) were scanned at A254 nm, and the diagram highlights the position of the 40S, 60S, 80S, and polyribosomal subunits. The dotted line highlights the reproducible reduction in the A254 signal in eluates from the IFITM1/IFITM3 null cells. (**B**) Following fractionation of material from (**A**), samples from fractions 1–18 were precipitated by trichloroacetic acid, and they were analyzed for IFITM1/3 protein expression (enriched in all wt-SiHa fractions). (**E**) Quantitation of the ratio of 40S, 60S, and 80S A254 nm peaks in wt-SiHa cells and IFNγ-stimulated IFITM1/IFITM3 null cells.

## Data Availability

The mass spectrometry (MS) proteomics data have been deposited to the ProteomeXchange Consortium via the PRIDE partner repository with the dataset identifier PXD030540, Username: reviewer_pxd030540@ebi.ac.uk, Password: jM0wyPd7. The entire set of RNA-seq files have been deposited to the datadryad.org with DOI https://doi.org/10.5061/dryad.c59zw3r92.

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
