# Peer review of "Emergent Role of IFITM1/3 towards Splicing Factor (SRSF1) and Antigen-Presenting Molecule (HLA-B) in Cervical Cancer"

_biomolecules, 2022, doi:10.3390/biom12081090_

Round 1

Reviewer 1 Report

The study by Gomez-Herranz identify novel IFITM1/3 interacting proteins including splicing regulators like SRSFs and U2AF1. IFNγ treatment increases IFITM1/3-SRSF1 foci in the SiHa cells, however interestingly SRSF1 protein level do not alter in WT vs IFITM1/3 KO cells. Further authors identified that HAL-B mRNA level don't change much but protein level drops in IFITM1/3 KO cells compared to WT after IFNγ treatment and reasoned that IFITM1/3 regulates HAL-B level by binding with mRNA. However, I have several concerns though this manuscript provides several important information specially the the conclusion drawn based on the experiments. This study lacks IFITM1/3 cytosolic and nuclear interacting proteins and thats why lack of clear picture of this study specially the mechanistic point of view.

1. Easiest experiment could be the cell fractionation (cytosolic and nuclear) and check expression and distribution of HAL-B, IFITM1/3, and SRSF1 at mRNA and protein level.

2. It may be possible that IFN treatment results into IFITM1/3  localize into late endosome or lysosome with HLA-B, which can be addressed or ruled out by co-immunostaining or protein-RNA staining.

3. CLIP is the best method to identify RNA binding proteins, why authors don't use this method?

4. For all microscopy- antibody and probe staining should be in separate panels followed by merged, so one can easily understand the position and localization?

5. Why 80S ribosomal fraction reduces in IFITM1/3 KO cells compared to WT.

6. IFITM1/3 binds to HLA-B mRNA- which region of mRNA? this is very important to explain regulatory behavior of IFITM1/3.

7. Can author measure mRNA and protein level of HLA-B at 0, 4, 6, 24, 48, and 72h after IFN treatment and how they are correlated?

8. Statistics are missing in all the RT-qPCR graph?

9. Author identify SRSF1 and RPL7a but how they help IFITM1/3 to regulate HLA-B expression is missing.

Reviewer 2 Report

In this manuscript, Maria Gomez-Herranz et a. investigated protein-protein interactions for IFITM1/3 in the context of cancer to clarify how IFITM1/3 attenuates the expression of targeted proteins such as HLA-B, and found that IFITM1/3 interacted with HLA-B mRNA in response to IFNγ stimulation and HLA-B gene expression at mRNA level does not account for lowered HLA-B protein synthesis in response to IFNγ. The authors concluded that changes in HLA-B expression could impact the presentation and recognition of oncogenic antigens on the cell surface by cytotoxic T cells and, ultimately, limit tumor cell eradication. I think this is a thought-provoking study and has very interesting results. However, direct evidence has not been adequately presented, and this manuscript has various concerns. Therefore, the authors must improve their manuscript according to the suggestions.

Major comments

1. The authors claimed protein-protein interaction and protein-RNA interaction using Figures 1, 2 and 4. Immunohistochemistry results certainly indicate that they are localized in similar locations. However, co-localize does not mean that they are interacting with each other. Using pull-down experiments, SPR, etc., the authors should provide direct evidence of an interaction.

2. SRSF1 shows nuclear speckle localization, but the low magnification in the presented figures makes it difficult to confirm. More magnified figures should be shown. Also, related to the above, if the authors want to show co-localization, it would be easier to understand if each antibody is labeled differently. Why use the same FITC?

3. The authors also claimed the association between SRSF1 protein and HLA-B RNA. If so, how is the splicing of the HLA-B gene altered? Please provide the results of HLA-B mRNA expression analysis.

Minor comments

1. Abbreviations should be spelled out on the first appearance.

2. It is incongruous that Figures 3 and 5 are tucked in the middle of the Materials and Methods section. Please insert them near the mention in the results.

3. Why do the nuclei look so different in size in the histochemistry photos? For example, Figures 1A, B and E to F, Figure 2J and the others, Figure 4D and H.

Reviewer 3 Report

In this manuscript the authors investigate the mechanisms of HLA-B protein regulation by IFTM1/3 taking advantage of previously generated IFTM1/3 null cells. Through SILAC and LC-MS analysis authors found different IFITM1/3 protein interactors and focused on the association with SRSF1. They exclude that IFITM1/3 could act by altering transcript expression by qPCR and RNA-seq analysis and found instead an involvement of IFITM1/3 in protein expression.

Since IFITMs are associated with cancer progression and with the cellular viral response the presented article is of relevance to the field and may open new perspectives in the study of recognition and presentation of oncogenic antigens by immune cells. Moreover, the presented mechanism for HLA-B protein could be a more general way of regulation by IFTM1/3 proteins.

While this article is well written and designed and a detailed description of methods is present there are a few queries that need to be clarified

Major general concept comments

1. Authors state IFTM1/3 are the main player in the formation of IFNγ induced IFTMs/SRSF1 foci since SRSF1 expression is not affected in IFTM1/3 null cells and by the treatment. However it is well known that phosphorylation status of SRSF1 can differentially affect several properties of the proteins (i.e. protein-protein interactions).

Authors should control SRSF1 phosphorylation status and SRPK1 (SRSF1 specific kinase) expression levels in wild type and IFTM1/3 null cells and untreated and IFNγ-treated cells.

2. Authors combine figure 1 and 2 to conclude that IFNγ induces IFTM1/3-SRSF1-HLA-B RNA interactions. What is missing is rISH-PLA with SRSF1 antibody to validate SRSF1-HLA-B interaction.

Or at least the authors should search for putative binding sites for SRSF1 in HLA-B mRNA sequence or mining public available CLIP-Seq data to support SRSF1 direct binding.

3. To consolidate their mechanism the authors could transiently knockdown SRSF1 in wild-type and IFTM1/3 null cells and assess HLA-B protein expression to understand to which extent IFITM1/3 is necessary for SRSF1 action on HLA-B mRNA.

Minor specific comments

For clarity, figures 3 and 5 in the Methods section should be included in the main text.

In Figures 1, 2 and 4 specify the used microscope objectives.

In Figure 2K count of foci/cell in "No probe" sample doesn’t correspond to figure 2A where no foci at all are visible. Is there a problem with the quantification? If not can the authors use a more representative image in panel A?

Gene names in figure 3B are too short to be read, please increase the font.

Sentence 496-498, “Moreover, there is a reduction of IFITM1 and IFITM3 mRNA abundance in IFNγ-stimulated IFITM1/IFITM3 null cells compared to wt-SiHa cells (Fig 3A III and 3 IV, respectively)” is not clear since the reduction is seen in IFITM1/IFITM3 null compared to wt-SiHa cells independently from IFNγ. Instead, treatment with IFNγ increases the expression levels compared to non-treated cells.

Lane 666 In the section of “supporting information” the title of Figure S1 doesn’t correspond to its content (CRISPR/Cas9 scheme is not present).

Round 2

Reviewer 1 Report

I agree with the explanation and concerns raised by the authors however, few supporting experiments could make this manuscript logically stronger and clear to the readers of Biomolecules.

Reviewer 2 Report

The authors responded as “we are very interested in understanding the potential role between SRSF1 protein and HLA-B RNA” to my comment that request results of HLA-B mRNA expression analysis. I think the regulation of splicing is the physiological role of the SRSF1, so I'll concede their point, although I'm not happy about it. The authors responded to reviewers' comments as appropriate, and the quality of the paper improved. I find this manuscript acceptable.

Reviewer 3 Report

Thank you for your revised version. Most of my requests have been met, the revised paper has strengthened the data. Now the paper is acceptable.